# Exploring the Impact of Chitosan Composites as Artificial Organs

**DOI:** 10.3390/polym14081587

**Published:** 2022-04-13

**Authors:** Iyyakkannu Sivanesan, Nazim Hasan, Manikandan Muthu, Gowsalya Blessing, Judy Gopal, Sechul Chun, Juhyun Shin, Jae-Wook Oh

**Affiliations:** 1Department of Bioresources and Food Science, Institute of Natural Science and Agriculture, Konkuk University, 1 Hwayang-dong, Gwangjin-gu, Seoul 05029, Korea; siva74@konkuk.ac.kr (I.S.); scchun@konkuk.ac.kr (S.C.); 2Department of Chemistry, Faculty of Science, Jazan University, Jazan P. O. Box 114, Saudi Arabia; hhasan@jazanu.edu.sa; 3Division of Research and Innovation, Department of Biotechnology, Saveetha School of Engineering, Saveetha Institute of Medical and Technical Sciences (SIMATS), Thandalam, Chennai 602105, Tamil Nadu, India; bhagatmani@gmail.com; 4Laboratory of Neo Natural Farming, Chunnampet 603401, Tamil Nadu, India; gowsalyablessing@gmail.com (G.B.); jejudy777@gmail.com (J.G.); 5Department of Stem Cell and Regenerative Biotechnology, Konkuk University, 1 Hwayang-dong, Gwangjin-gu, Seoul 05029, Korea; junejhs@konkuk.ac.kr

**Keywords:** chitosan, biomedical applications, artificial organs, composites, tissue regeneration

## Abstract

Chitosan and its allies have in multiple ways expanded into the medical, food, chemical, and biological industries and is still expanding. With its humble beginnings from marine shell wastes, the deacetylated form of chitin has come a long way in clinical practices. The biomedical applications of chitosan are truly a feather on its cap, with rarer aspects being chitosan’s role in tissue regeneration and artificial organs. Tissue regeneration is a highly advanced and sensitive biomedical application, and the very fact that chitosan is premiering here is an authentication of its ability to deliver. In this review, the various biomedical applications of chitosan are touched on briefly. The synthesis methodologies that are specific for tissue engineering and biomedical applications have been listed. What has been achieved using chitosan and chitosan composites in artificial organ research as well as tissue regeneration has been surveyed and presented. The lack of enthusiasm, as demonstrated by the very few reports online with respect to chitosan composites and artificial organs, is highlighted, and the reasons for this lapse speculated. What more needs be done to expand chitosan and its allies for a better utilization and exploitation to best benefit the construction of artificial organs and building of tissue analogs has been discussed.

## 1. Introduction

Chitosan is composed of D-glucosamine and N-acetyl-D-glucosamine that are randomly β-(1-4)-linked. It is obtained through the deacetylation of chitin. Chitin is the second most abundant polysaccharide in nature after cellulose. The main source of chitin is the exoskeleton of crustaceans; other insects and microorganisms are alternate sources [1,2,3]. The deacetylation degree (DD) of chitosan indicates the number of amino groups along the chain. To be classified ‘‘chitosan’’, the deacetylated chitin should contain >60% of D-glucosamine residues [4,5].

The deacetylation of chitin is conducted chemically or biologically [6,7]. The chemical method is commonly used at an industrial scale, with strong acids and alkalis [8]. Biological methods use microorganisms and enzymes [9]. For biomedical applications, chitosan purity, crystallinity, molecular weight (Mw), and deacetylation degree (DD) are important [10]. These factors deeply correlate with chitosan’s mechanical and biological properties [11].

The increasing interest in chitosan as a biomaterial is due to its excellent biocompatibility, non-allergenicity, non-toxicity, and biodegradability. Its antifungal, antitumor, antibacterial, antioxidant, anti-inflammatory, immunoadjuvant, anti-thrombogenic, and anti-cholesteremic attributes are also crucial when it comes to biomedical applications [12,13]. It possesses high versatility and is processable in diverse morphologies, such as fibers (nanofibers), films, sponges, gels, beads, particles (and nanoparticles), membranes, and scaffolds [14,15]. All these properties render chitosan worthy of biomedical and pharmaceutical operations, such as drug delivery, gene delivery, tissue engineering, regenerative therapies, and others.

The presence of the protonable amino group in chitosan allows for its biomedical properties. The mucoadhesion of chitosan is because of the negatively charged sialic acid residues in the mucin—the glycoprotein that composes the mucus. Mucoadhesion is directly related to the DD of chitosan: if chitosan DD increases, the number of positive charges also increases, which leads to improved mucoadhesive properties [16]. The haemostatic activity of chitosan is due to positive charges on the chitosan backbone. The negatively charged red blood cells interact with the positively charged chitosan [17,18,19]. Due to its positive charges, chitosan can also interact with the negative part of cell membranes, resulting in reorganization of the tight junction proteins, and it is enhanced. Chitosan is actually more cytocompatible in vitro than chitin.

Considering all the above properties, it is not strange that chitosan is useful for biomedical and pharmaceutical applications, especially for sutures, dental and bone implants, and as artificial skin [3]. Within this framework, chitosan has shown biocompatibility [11,20] and is approved by the Food and Drug Administration (FDA) for use in wound dressings [21,22].

This review overviews the various biomedical applications of chitosan and the synthesis of biomedical chitosan. The use of chitosan for construction of artificial organs is elaborately dealt with, and the milestones achieved have been outlined. This is the first review that consolidates the published literature related to chitosan composites and artificial organs. The future of chitosan and the need to expand its application in the area of artificial organs is emphasized. Previous reviews mostly focused on the biomedical aspects and the tissue-engineering aspects of chitosan composites; this review specifically highlights the status quo of chitosan in the area of artificial organ generation.

## 2. Snapshot of Biomedical Achievements of Chitosan

Biomedical applications of chitosan are ideally based on its biocompatibility and biodegradability. The biomedical achievements of chitosan have been extensively reviewed by various authors [23,24,25,26]. Chitosan and its derivatives have numerous beneficial effects, including anti-Alzheimer’s disease, anti-atherosclerosis, anti-inflammatory diseases, kidney injury alleviation, improvement of intestinal barrier function, enhancement of beneficial gut microbiota, alleviation of obesity and type II diabetes, prevention of inflammatory diseases, anticancer activity, antiaging activity, and antibacterial activity and in addition serve as valuable biomaterials for wound dressing and drug delivery. Herein, a brief overview of the various aspects of the biomedical applications of chitosan are enlisted.

Tissue engineering requires a supporting natural and synthetic polymer material. Polylactic acid (PLA) and polyglycolic acid (PGA) are usually used for fibroblasts growth at laboratory scale [27,28]. Chitosan has now become one of the leading cell culture media because of its biocompatibility and the reason that it accelerates the growth process. PLA nanofibers with chitosan coating have been used to grow osteoblasts and promote regeneration of bone tissue. The biodegradability, biocompatibility, antibacterial activity, and low immunogenicity of chitosan and its derivatives come in handy for wound-healing applications [29]. They provide three-dimensional tissue growth matrices, activate macrophages, and stimulate cell proliferation [30]. Moreover, chitosan facilitates pronuclear leukocytes and activates macrophage fibroblasts, enhancing granulation leading to tissue repair [31]. Slow degradation of N-acetyl-β-D-glucosamine activates fibroblast proliferation, collagen precipitation, and the subsequent synthesis of hyaluronic acid, accelerating wound healing and preventing scarring [32]. Sankar et al. [33] reported a lyophilized glutaraldehyde crosslinked chitosan sponge for blood hemostasis. Composite tricalcium phosphate-chitosan, has been demonstrated as a bone substitute and as a tissue-engineering scaffold.

Chitosan nanoparticles have been used as bone substitutes, improving drug release capacity and serving as osteoblast cell culture scaffolds [34]. Chitosan is reported for colon-targeted delivery of a drug due to its pH sensitivity and its complete digestion by the colonic bacteria [35,36]. Chitosan tripolyphosphate (TPP) nanoparticles can adhere to and help retain drugs, such as doxorubicin on mucosal surfaces [37]. The 5-fluorouracil-based chitosan nanoparticles are well known for their tumor-related reports [38]. Insulin-based chitosan nanoparticles exhibit high drug entrapment efficiency, good stabilization, low outbreak, and steady release of insulin [39]. Chitosan crosslinked with poly (ethylene glycol) dialdehyde, formed a hydrogel that increases protein release [40]. Oral delivery of glucomannan-modified chitosan nanoparticles was studied on mice [41,42,43,44].

The antibody (IgA)-based chitosan-dextran sulfate nanoparticles with pertussis toxin showed prior absorption of IgA-based chitosan-dextran sulfate nanoparticles through nasal membranes or M cells in mice following intranasal immunization in vivo [45]. N-2-HACC and N,O-carboxymethyl chitosan (CMC) nanoparticles have been synthesized and evaluated as vaccine adjuvants for the Newcastle disease vaccine (NDV) and infectious bronchitis vaccine (IBV). The immune responses in chickens revealed that those nanoparticles containing NDV/IBV can induce better intranasal inoculation of IgG and IgA antibodies and enhance the proliferation of lymphocytes [46].

Chitosan nanoparticles combined with plasmid DNA enhances antigen-specific immunity [47]. Plasmid DNA encapsulated chitosan nanoparticles resulted in effectively enclosed plasmid DNA and its in vivo expression [48]. Chitosan nanoparticles loaded with IL-2 expression plasmids have been demonstrated for gene-based immune therapy and have been confirmed to deliver plasmid unchanged during encapsulation [49,50]. Chitosan is a cationic polyelectrolyte; its positive charge strongly and electrostatically interacts with negatively charged DNA and protects it from nuclease degradation [36,37]. Nanoparticles of folic acid-chitosan-DNA have been used in gene therapy [38]. Deoxycholic acid-modified chitosan were used as a delivery carrier for the transfection of genetic material in mammalian cells [39]. Chitosan-alginate core shell nanoparticles encapsulated with enhanced green fluorescent protein (EGFP)-encoded plasmids were endocytosed by NIH 3T3 cells, causing swelling of transport vesicles which renders gene escaping before entering digestive endolysosomal compartments, promoting the gene transfection rate [40,41].

Chitosan membranes have been used in wound healing with potential applications in patients with deep burns, wounds, etc. PVA–chitosan membranes have enhanced mechanical properties [51]. Chitosan films with oleic acid and glycerol at 1% have been used in vivo in Wistar^®^ rats. The results showed that implanted chitosan films were biocompatible and bioabsorbable and have wound-healing prospects [52]. Silver combined with chitosan has been used as an antibacterial agent to prevent wound infections [6,53]. Chitosan has been considered a natural antioxidant [54,55,56]. Functionalization of chitosan with epigallocatechin gallate (EGCG) affects its antimicrobial and antioxidant activities [57]. Chitosan antioxidant activity is increased by grafting it with protocatechuic acid, which is a natural phenolic antioxidant [58]. Chitosan modified with monomethyl fumaric acid provided it with antibacterial activity against Gram-positive and Gram-negative bacteria [59]. Other chitosan derivatives with antimicrobial activity are chitosan modified with thioglycolic acid [60] and chitosan with polyethylene glycol diacrylate (PEGDA) [61].

## 3. Development of Chitosan-Based Materials for Biomedical Applications

Chitosan as a biomaterial inherently possesses an architectural similarity with native extracellular matrix and is able to ensure cell growth and regeneration. Over the years, different biomaterials with suitable characteristic properties for tissue engineering were explored. When it comes to the synthesis of chitosan biomaterials, solvent casting, the compression-molding method, freeze drying, electrospinning, microwave assisted, and dense-gas-foaming methods are the popularly used methods [62]. In the cases of biomaterial applications, the synthesis procedures are highly specific.

### 3.1. Solvent-Casting/Solvent-Evaporation/Tape-Casting Method

The solvent-casting/solvent-evaporation/tape-casting method is used to obtain chitosan membranes with ultrafine pores. The polymer in the acidic solution is stirred overnight. The homogenized solution is filtered and oven dried. The dried membranes can be peeled off from the petri plate and used [63]. Chitosan in lactic acid is more flexible and transparent. The addition of plasticizers (glycerol, polyethylene glycol, and sorbitol) improves the tensile strength of the membranes [64,65,66].

### 3.2. Compression-Molding Method

The compression-molding method produces chitosan in membrane form. This method requires preconditioning by drying chitosan and choline chloride in a drying oven. The dried powder is finely ground for at least 15 min and oven dried at 70 °C before treating with aqueous acetic solution. A hydraulic tool is used to form membranes [67]. This method is deployed at lab scale; it is not useful for large scale processes [68].

### 3.3. Freeze-Drying Method

The freeze-drying process involves low temperature dehydration where the sample is frozen and the pressure lowered and ice removed. Freeze drying results in a high-quality product as the original structure of the sample is not altered at low temperature. The freeze-drying method has been applied to prepare porous scaffolds (hydrogel and sponge material) [69,70]. Slow drying results in a porous structure [71,72,73,74]. The freeze-drying method is a promising application in preparation of biomaterials, as it does not require a high temperature, laborious procedures, and expensive machinery.

### 3.4. Electrospinning Method

The electrospinning technique produces ultrafine, nanometer-sized fibers that can organize into structures mimicking the native extracellular matrix, which can facilitate cell adhesion and growth similar to the natural system [75,76]. Chitosan scaffolds are prepared via the electrospinning technique by solubilizing chitosan in an aqueous acetic or formic acid solution, homogenously blending with other polymer(s), and filling into a syringe. Under high voltage, a fibrous non-woven web is detached from the cylindrical mandrel and from the nozzle forming a jet. The nanofibers were air dried at 37 °C overnight and sterilized with UV radiation prior to use [77]. Cross-linkers such as genipin, glutathione, and glutaraldehyde are also used with electrospun fibers of chitosan [77].

### 3.5. Microwave-Assisted Method

The microwave assisted technique utilized microwave irradiation to generate cross-linked polymeric chains. This method has been used to prepare chitosan-based hydrogel systems for tissue engineering. The hydrogels possess chains interlink with each other and need no chemical cross-linker, making it less toxic to biological cells [78,79,80].

### 3.6. Dense-Gas-Foaming Method

The dense-gas-foaming method synthesizes highly porous hydrogel systems using supercritical gas. A polymeric solution is injected into Teflon molds of high-pressure vessels. The vessel is stabilized to attain thermal equilibrium, and then optimized carbon dioxide (CO_2_) gas is pressurized for a specific period of time. The pressure is released, and the sample is collected. Prepared hydrogel is washed again and stirred in PBS for later use [81]. Supercritical CO_2_ results in uniform-sized pores in the polymer matrix system [82].

## 4. Chitosan-Based Artificial Organs

Ever since Dr. Anthony Atala pioneered regenerative medicine, artificial organ generation has come a long way. Chitosan and its composites have emerged in diverse morphologies and forms and have significantly impacted a crucial area of human physiology via replacement of impaired portions. Figure 1 gives the listed chitosan forms and the areas of the human body they have impacted.

### 4.1. Chitosan as Artificial Membranes

Hirano et al. prepared membranes from chitin and its derivatives to improve the dialysis-related properties of membranes [83,84]. The N-acetyl chitosan membranes exhibit enhanced permeability with similar properties to that of the traditional Amicon Diaflo membrane UM 10. Proteins have been incorporated in the chitosan membranes to prepare chitosan-blended membranes [85]. The protein-blended membranes compared to standard membranes or unblended chitosan [86] exhibited improved permeability to urea, creatinine, uric acid, and glucose.

Bioactive complex immobilized albumin-blended chitosan membranes are also reported. Hirano et al., coated silk filament sutures with chitosan, N-acetyl chitosan, or N-hexanoyl chitosan and evaluated in vivo blood compatibility properties [87]. They observed that the chitosan membrane was thrombogenic, but N-acetyl and N-hexanoyl chitosan membranes were nonthrombogenic. Enhanced blood compatibility of chitosan membranes through various surface modifications to reduce interferences is reported. Bioactive molecules were immobilized on liposome-modified, albumin-blended chitosan membranes, made nonthrombogenic by immobilizing bioactive molecules, such as PGE1, Hirudin, Heparin, or AT-111, via the carbodiimide functional moiety [88]. Such novel biomembranes demonstrated better permeability towards small molecules and showed a dramatic reduction in platelet attachment. Phosphoryl choline bilayer immobilized on albumin-blended chitosan membranes improve their blood compatibility [89]. Such phospholipid bilayer membranes reduced surface platelet adherence and retained their excellent permeability for small molecules. Chondroitin sulphate and phosphoryl choline immobilized on chitosan with albumin are also reported. The permeability of various molecules through different biomolecules immobilized on albuminated chitosan membranes during a 4 h dialysis cycle is reported. Therefore, these modified membranes may have wider applications towards hemodialysis by improving permeability and blood compatibility.

### 4.2. Chitosan as Artificial Skin

Kim and Min fabricated a wound-covering material from polyelectrolyte complexes of chitosan with sulfonated chitosan [90]. Wound healing is expected to be accelerated by oligomers of degraded chitosan by tissue enzymes leading to regeneration of skin tissue in the wound areas. A chitosan collagen blend in the ratio of 7:3 was used by the authors to treat wounds in male rabbits. Bare chitosan and chitosan coated with a dense layer of fibrin gel were also used for comparison; all the chitosan-grafted animals showed similar healing patterns.

The healing of a skin wound involves cellular, molecular, physiological, and biological processes. Immediate coverage of the wound with wound dressing is the cornerstone of wound management. Numerous studies report the use of chitosan as a skin substitute material in skin tissue engineering. Chitosan stands advantageous when used for wound healing because it impacts hemostasis, accelerating tissue regeneration and stimulates fibroblast synthesis of collagen [91,92,93,94]. Ueno et al. (1999) demonstrated that chitosan in the form of chitosan cotton accelerates wound healing by promoting infiltration of poly morphonuclear (PMN) cells at the wound site, which is crucial for rapid wound healing [95]. Recently, Mizuno et al. (2003) demonstrated chitosan as an ideal wound-healing material [96]. Howling et al. (2001) demonstrated that highly deacetylated chitosan are more biologically active than chitin [97]. Yan et al. (2000) reported chitosan-blended alginate as polyelectrolyte complex (PEC) membranes [98]. These biodegradable chitosan–alginate PEC membranes showed greater stability to pH changes and were more effective as controlled-release membranes than chitosan or alginate itself [99]. PEC membranes accelerated healing of incisional wounds in rat models. Ma et al. (2003) fabricated porous chitosan/collagen scaffolds by cross-linking with glutaraldehyde and freeze-drying to improve biostability and good biocompatibility [100].

### 4.3. Chitosan as Artificial Bone

In bone tissue engineering, the biodegradable substitutes take the position as a temporary skeleton inserted into the defective sites of the lost bone sites to support and stimulate bone tissue regeneration while they gradually degrade and get replaced by new bone tissue. Both bioactive ceramics and polymers have been developed in this direction and employed as tissue-engineering scaffolds. Bioactive ceramics being chemically similar to natural bone, allow osteogenesis [101]. However, brittleness and low biodegradability restrict their clinical applications; it is from this perspective that natural and synthetic polymers have been considered as bone substitutes. Chitosan especially has been extensively used in bone tissue engineering, owing to its capacity to promote growth and mineral rich matrix deposition by osteoblasts. Moreover, it is biocompatible and can be molded into porous structures (allows osteoconduction) [102]. Several studies have focused on chitosan–calcium phosphates (CP) composites for this purpose in bone tissue engineering. Beta-tricalcium phosphate (β-TCP) and hydroxyapatite (HA) of CP bioceramics are excellent candidates for bone repair and regeneration because of their similarity in chemical composition with inorganic components of bone. Zhang et al. prepared CP bioceramics embedded in chitosan sponge to enhance the mechanical property of the ceramic phase via matrix reinforcement for preserving the osteoblast phenotype [103,104].

### 4.4. Chitosan as Artificial Cartilage

Tissue engineering of articular cartilage begins with the isolation of articular chondrocytes (precursor cells), followed by seeding into a biocompatible matrix or scaffold for cultivation and subsequent implantation into the joint [105] When it comes to cartilage repair, the choice of biomaterial is very important [106]. Cartilage-specific extracellular matrix (ECM) components, such as type II collagen and GAGs, are known to play a crucial role [107,108]. The structural similarity of chitosan with various GAGs that are components of articular cartilage renders it as a perfect candidate for scaffolding material [105].

Lu et al. demonstrated that chitosan solution injected into the knee articular cavity of rats led to the significant increase in the density of chondrocytes in the knee articular cartilage, indicating that chitosan could be potentially beneficial to the wound healing of articular cartilage [109]. Mattioli-Belmonte et al. confirmed that the bone morphogenetic protein, (BMP)-7, associated with N,N dicarboxymethyl chitosan induces/facilitates repair of artificial cartilage lesions in rabbit [110]. The cationic nature of chitosan allows for the formation of ionic complexes of chitosan with the negatively charged GAGs. Sechriest et al. demonstrated biocompatibility and chondrogenic characteristics of GAG-augmented chitosan hydrogels [111,112]. Chitosan was also conjugated with hyaluronan to obtain a biomimetic matrix for chondrocytes, leading to enhanced synthesis of aggrecan and type II collagen and, similarly, to increase cellular adhesiveness of chitosan. Hsu et al. (2004) studied chitosan–alginate–hyaluronan scaffolds with or without covalent attachment with RGD-containing protein [113]. Cell-seeded scaffolds showed neocartilage formation in vitro and in rabbits. Lee et al. (2004) reported porous collagen/chitosan/GAG scaffolds loaded with transforming growth factor-*β*1(TGF-*β*1), which lead to controlled release of TGF-*β*1, promoting cartilage regeneration [114]. Moreover, the addition of chitosan to the collagen scaffold improved mechanical properties and stability of the collagen network [92]. Recently, Buschmann et al. showed that microfractured ovine defects are repaired with more hyaline cartilage when the defect is treated with in situ-solidified implants of chitosan–GP mixed with autologous whole blood compared to microfracture alone in an ovine model at six months [115]. Chitosan–GP/blood implants were applied to marrow-stimulated chondral defects in rabbit cartilage repair models [116] where they induced tissue repair [117,118] and hyaline repair [117,118].

### 4.5. Chitosan as Artificial Liver

Insufficient donor organs for orthotopic liver transplantation have increased the requirement for new therapies for acute and chronic liver disease [119]. Bioartificial liver (BAL) is a promising application of tissue engineering for the treatment of fulminant hepatic failure (FHF). Chitosan as a promising biomaterial has been applied towards liver tissue engineering due to its various properties. Chitosan is used as a scaffold for hepatocytes culture because of its structural similarity to GAGs (components of the liver CM) [120]. Chupa et al. demonstrated that chitosan and chitosan complexes with GAGs can modulate the activities of vascular endothelial and smooth muscle cells in vitro and in vivo [121].

Li et al. showed that the micro-structure of porous scaffolds enabled a large surface area for cell adherence to facilitate nutrient/oxygen transportation [122]. Wang et al. prepared a chitosan/collagen matrix (CCM) showing superior blood compatibility in implantable bioartificial liver (IBL) applications [123]. Chitosan modified with galactose residues improved hepatocyte attachment and maintained viability. Park et al. reported galactosylated chitosan (GC) as a new synthetic ECM for hepatocyte attachment [124]. Furthermore, Chung et al. suggested a potential ability to improve hepatocyte attachment to alginate (AL)/GC scaffolds for short-term cultures [125]. Li et al. conjugated fructose onto the porous chitosan scaffold, where fructose acts as a specific ligand of ASGPR in hepatocyte [122,124]. They showed that chitosan surfaces modified with fructose are able to induce cellular aggregates by enhancing liver-specific metabolic activities and cell density.

### 4.6. Chitosan as Artificial Nerve

More than any other form of trauma, nerve injuries are difficult to rectify because mature neurons (like many other cells in the body) have little capacity for replication. Once the nervous system is impaired, its recovery is difficult, leading to malfunctioning of other parts [126]. Chitosan has been seriously considered as an option for nerve regeneration due to its unique properties. Jianchun et al. reported that neurons cultured on the chitosan membrane grow well and that chitosan tubes greatly enhance the repair of the peripheral nervous system [127,128] and showed that chitosan fibers supported the adhesion, migration, and proliferation of SCs, which provide a similar guide for regenerating axons to Büngner bands in the nervous system [129]. Itoh et al. demonstrated the usability of hydroxyapatite-coated chitosan tubes (loaded with laminin-1 or laminin peptides) as scaffolds for peripheral nerve reconstruction [130]. Matsuda et al. immobilized laminin peptide in molecularly aligned chitosan by covalent bonding for nerve regeneration [131]. Kato et al. identified neurite-outgrowth-promoting sites on the human laminin α3 chain LG4module (A3G75 and A3G83) and prepared peptide-conjugated chitosan membranes [132]. Chávez-Delgado et al. confirmed that progesterone from chitosan prostheses provided better facial nerve regenerative response in rabbits [133]. Cao et al. further studied the physical, mechanical, and degradation properties of chitosan films and the affinity between Schwann cells SCs and the films [134]. Three kinds of cross-linked chitosan films were prepared with hexamethylene diisocyanate (HDI), epichlorohydrin (ECH), and glutaraldehyde (GA) as crosslinking agents, respectively. Crosslinking decreased the swelling degree and the degradation rate of the chitosan films, whereas it increased their hydrophilicity and elastic modulus. Other reports showed that chitosan blended with peptides lead to mechanical properties of scaffolds and could enhance nerve cell attachment. Mingyu et al. showed improved attachment, differentiation, and growth on the chitosan/poly(L-lysine) composite materials compared to cells cultured on chitosan membranes [135], owing to increased hydrophilicity. Cheng et al. added gelatin to chitosan to produce a soft and elastic complex that has good nerve cell affinity [136]. The chitosan/gelatin composite film showed a lower modulus and a higher percentage of elongation at break compared with chitosan film, and PC12 cells cultured on them differentiated more rapidly and extended longer neurites than on chitosan films. Frier et al. also developed chitin hydrogel tubes for use in this direction [137].

### 4.7. Chitosan as Artificial Blood Vessel

Vascular disease is the major reason for mortality in Western society [138]. Coronary artery and peripheral vascular disease are the leading causes of mortality, necessitating surgical interventions, including small-diameter bypass grafting with autologous veins or arteries. Commonly, vascular transplantation has been used for the treatment of vascular disease. Chupa et al. has made the effort to overcome both incomplete endothelization and smooth muscle cell hyperplasia through complexation of GAGs with porous chitosan scaffolds [133]. GAG-based materials hold promise because of their growth inhibitory effects on vascular smooth muscle cells and their anticoagulant activity. Chitosan as a scaffold of tissue-engineered blood vessels is sparsely reported. Madihally et al. fabricated a family of chitosan scaffolds, including heparin-modified porous tubes, which had the potential for application in blood vessel tissue engineering [5]. Kratz et al. (1997) also prepared insoluble ionic complexes with heparin [139]. The heparin–chitosan scaffolds showed excellent biocompatibility [140,141,142,143,144,145]. Heparin plays an important role in blood vessel tissue engineering [146]. The chitosan/heparin complex had good blood compatibility, and these scaffolds stimulated cell proliferation and initiated a thick, dense, and highly vascularized granulation layer [121,146]. Table 1 gives the summary of chitosan applications in the area of artificial organs.

## 5. Chitosan Composites in Organ 3D Bioprinting

Organ 3D bioprinting is an advanced 3D printing technology applied for assembling biomaterials in a layer-by-layer fashion to produce bioartificial organs that mimic their natural counterparts. Living cells and growth factors are encapsulated into the polymeric solutions or hydrogels to obtain 3D tissue and organ construction. Very few natural and synthetic polymers and their composites can be used as “bioinks” for tissue and organ 3D bioprinting. Chitosan molecules due to their biocompatibility, biodegradability, antibacterial activity, non-antigenicity, and bioadsorbility have become iconic polymers when it comes to organ 3D bioprinting. The solubility of chitosan in diluted acids, leading to protonated amino groups, enables chitosan to form complexes with metal ions, natural or synthetic anionic poly(acrylic acid) polymers, lipids, proteins, and deoxyribonucleic acid (DNA) [147,148,149]. The cationic feature of chitosan enables the formation of gel particles through electrostatic interactions. The hydrophilic structure of chitosan promotes cell adherence and proliferation on its scaffolds, rendering chitosan an ideal candidate for tissue repair and organ 3D bioprinting. The chitosan composites [150,151,152,153,154,155,156] that have been used as “bioinks” for organ 3D bioprinting have been described in detail by Li et al., 2019 [157].

## 6. Future Perspectives and Conclusions

Tissue engineering forms the fundamental basis for regeneration of injured body portions and to restore functions using laboratory-grown tissues or artificial implants and materials. For the construction of artificial organs, biomedical engineering uses living cells, signal molecules, and scaffolds. Chitosan is the most promising biomaterial in tissue engineering because of its dynamic physico-chemical and biological properties. This review surveyed the utility of chitosan for its potential to construct artificial tissues and organs. There are many challenges, such as poor mechanical property as an artificial substitute, effective delivery strategy of growth factors to chitosan-based scaffold, demonstrating biocompatibility, as well as sterility, that must be addressed in various implant applications. Figure 2 gives the overview of the various areas that chitosan and its composites have impacted. Given this fact, there is space for expansion and more sites to improvise.

Chitosan and tissue engineering are a popularly worked upon subject area as represented by the large number of publications (4836 papers) revealed by a pubmed search during 1981–2022 (Figure 3a) and reviews [25,26,27,28,158,159].

Chitosan and artificial organs as keywords on pubmed hit a 239 mark, which is a huge step down (Figure 3b). The aspect reviewed here, namely, chitosan composites and artificial organs, is addressed by merely a handful of reports (42 papers), where most of them are research papers (Figure 3c). The present review is one of the couple of reviews on this subject area. As represented in the graph in Figure 3c, there is a significant lack of enthusiasm in the area of chitosan composites and artificial organs. While chitosan has proved effective in tissue culture, the more advanced and state-of-the-art form of chitosan, which are its composites and derivatives, not being practically applied is rather strangely interesting. Chitosan composites broke most of the limitations of chitosan; there is no doubt that they have a lot to offer. This review hopes to create an awareness that could spark interesting applications and ensuing reports in this subject area. Speculating the reason why chitosan composites have been scarcely attempted, it appears that the interdisciplinary nature of the research could be the limiting factor. The chitosan composites are synthesized by chemists and need to be successfully transferred to tissue engineers to be applied in real time to artificial organ engineering. Collaborative working is the basis of successful practical implementation. The lack of which could be the reason for the break in the flow of the synthesized products to the application platforms.

Moving forward, down the course of the review, it was observed that the toxicity aspects of chitosan have not been clearly worked out. Any biomaterial, especially those considered for artificial organ-based applications and those meant to stay in the body for a long time, the toxicity aspect needs to have intense clarity. Moreover, chitosan derivatives nor chitosan nanoparticles, which are the novel forms of chitosan, have not been applied to the area of artificial organs, while these have been highly applied to various other biomedical aspects. This is a direction that needs more research focus. Chitosan is a high potential resource; delving deep to harness the various modifications and derivatives available will lead to eventual expansion into unexplored areas of artificial organs.

This review showed that chitosan has a huge potential in artificial organ regeneration research and development through chitosan and its composites. It is a versatile application area that has much scope for expansion, provided the more resourceful chitosan composites and chitosan derivatives are appropriately implemented into artificial organ development.

## Figures and Tables

**Figure 1 polymers-14-01587-f001:**
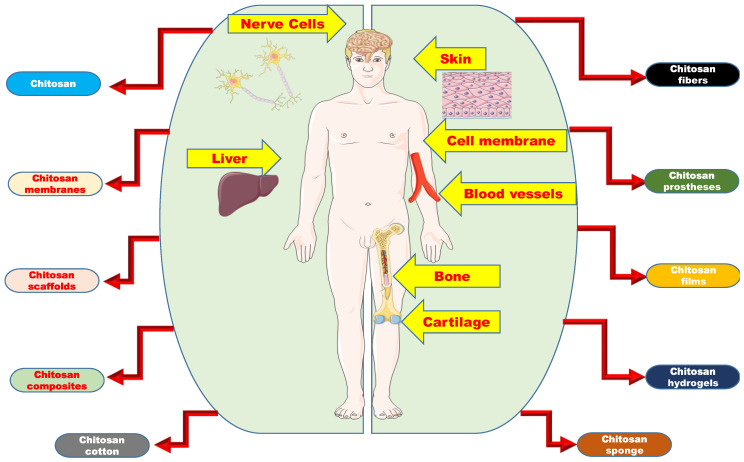
Overview of the organs to which chitosan and its composites have contributed to and the various chitosan forms that have gone into the making of the artificial organs.

**Figure 2 polymers-14-01587-f002:**
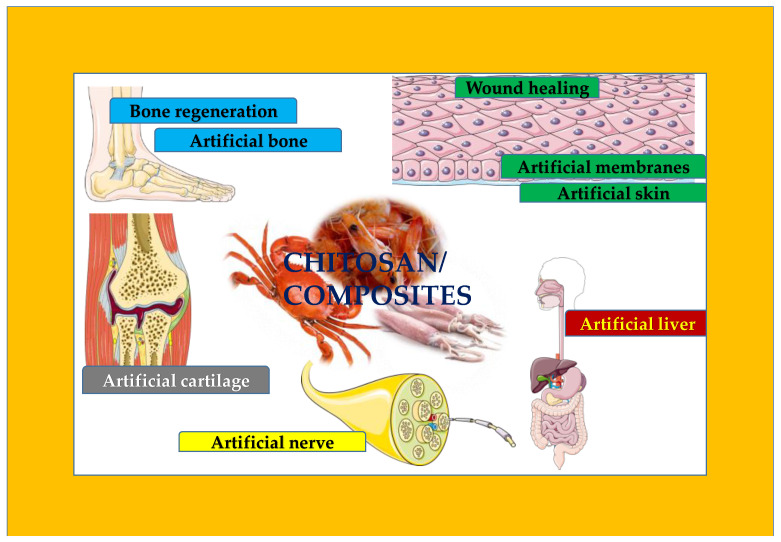
Overview of the areas impacted by chitosan and its composites.

**Figure 3 polymers-14-01587-f003:**
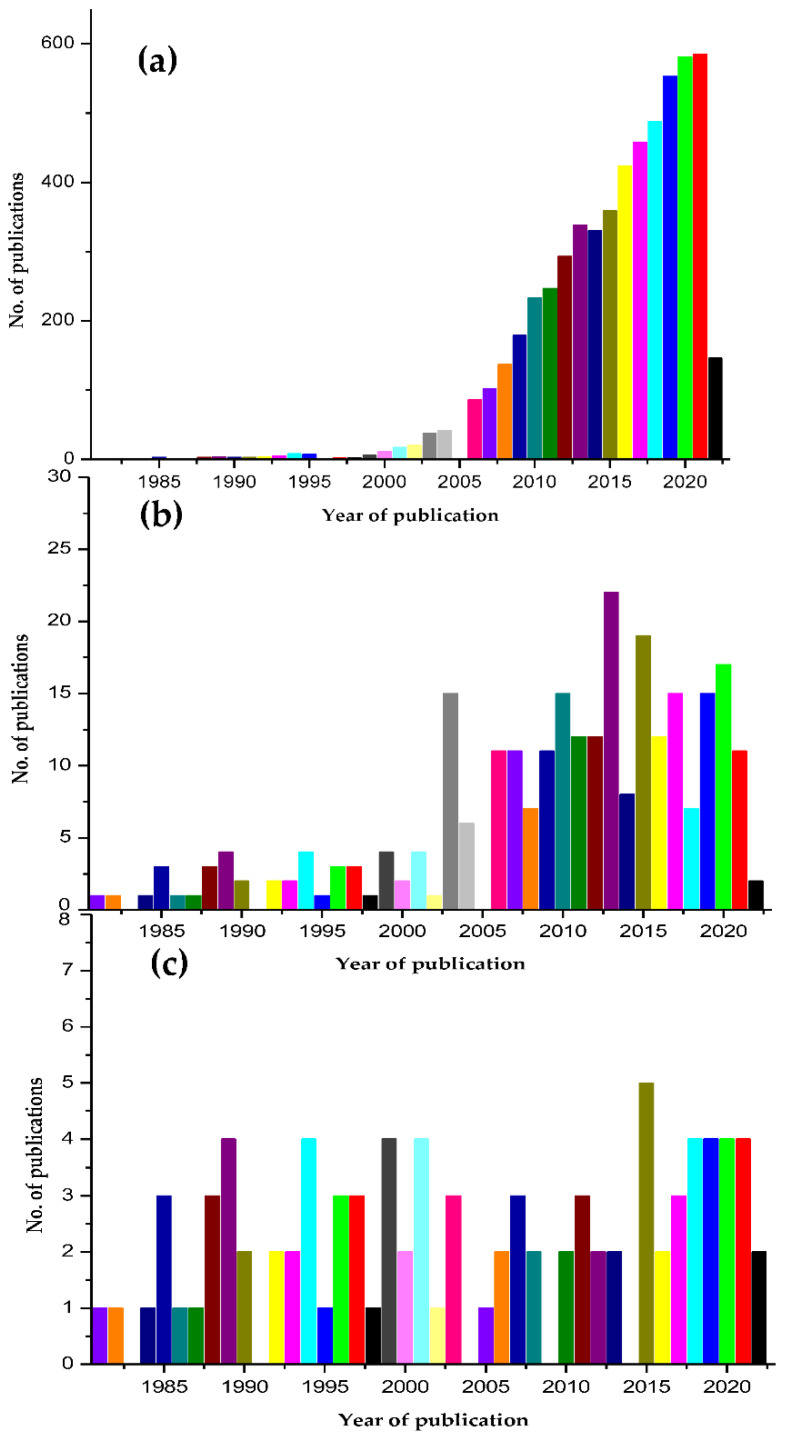
Comparative graph depicting the dearth in publications in the area of chitosan composites and artificial organs based on a pubmed search during the period of 1981–2022. Graph based on pubmed search with keywords (**a**) chitosan and tissue engineering (**b**) chitosan and artificial organs (**c**) chitosan composites and artificial organs.

**Table 1 polymers-14-01587-t001:** Summary of applications of chitosan and composites towards artificial organs.

Chitosan/Chitosan Derivative	Artificial Organ	Specified Application	Enhanced Properties	References
N-acetyl chitosan	Artificial membranes	Dialysis membrane	Improved dialysis membrane, permeability similar to Amicon Diaflo membrane	[85]
Chitosan:protein	Artificial membranes	Dialysis membrane	Improved permeability to urea, creatinine, uric acid, and glucose	[86]
N-acetyl and N-hexanoyl chitosan membranes	Artificial membranes	Dialysis membrane	Nonthrombogenic membrane properties	[87]
Bioactive-complex-immobilized, albumin-blended chitosan membranes	Artificial membranes	Dialysis membrane	Liposome-modified membranes were nonthrombogenic, better permeability, reduced platelet attachment.	[88]
Phosphoryl choline bilayer immobilized on albumin-blended chitosan membranes	Artificial membranes	Dialysis membrane	Similar permeability as chitosan membranes and drastic reduction in platelet adhesion offering improved permeability and blood compatibility	[89]
Polyelectrolyte complexes of chitosan with sulfonated chitosan	Artificial skin	Wound healing	Regeneration of skin in wound area, promoting wound healing	[90]
Chitosan	Artificial skin	Skin tissue engineering	Hemostasis, accelerating the tissue regeneration, and stimulating the fibroblast synthesis of collagen	[91,92,93]
Chitosan cotton	Artificial skin	Wound healing	Promotes infiltration of PMN cells at the wound site	[94]
Chitosan	Artificial skin	Wound healing	Promotes basic fibroblast growth factor(bFGF), accelerated healing	[95]
Highly deacetylated chitosan	Artificial skin	Wound healing	Rapid wound healing	[96]
Chitosan–alginate PEC membranes	Controlled release membranes	Wound healing	Accelerated healing of incisional wounds in a rat model	[97,98]
Porous chitosan/collagen scaffold by cross-linking with glutaraldehyde	Artificial skin	Wound healing	Good biocompatibility and induces fibroblasts infiltration	[99]
Chitosan–calcium phosphates (CP)	Artificial bone	Bone tissue engineering	Minimizes inflammation, biocompatible, biodegradable, moldable, porous allowing osteoconduction	[100,101]
CP embedded with chitosan sponge	Artificial bone	Bone reinforcement	Enhanced mechanical properties, matrix reinforcement, preserving osteoblast phenotype	[102,103]
Chitosan injected into the rat’s knee articular cavity	Artificial cartilage	Wound healing of articular cartilage	Significant increase in the density of chondrocytes in the knee articular cartilage	[109]
Bone morphogenetic protein (BMP)-7/N,N dicarboxymethyl chitosan	Artificial cartilage	Repair	Artificial cartilage lesions in rabbit	[110]
GAG-augmented chitosan hydrogel surfaces	Artificial cartilage	Bone tissue engineering	biocompatibility and the chondrogenic characteristics	[111,112,113]
Chitosan–alginate–hyaluronan scaffolds	Artificial cartilage	repair	Showed neocartilage formation in vitro	[114]
Porous collagen/chitosan/GAG scaffolds with transforming growth factor-*β*1(TGF-*β*1)	Artificial cartilage	Bone tissue engineering	Controlled release of TGF-*β*1 and promoted cartilage regeneration	[115]
In situ-solidified chitosan–GP blood implants	Artificial cartilage	Microfractured ovine defect repair	Chitosan is thrombogenic and actively stimulates the wound repair process	[116,117]
Chitosan/collagen matrix (CCM) by cross-linking agent EDC	Artificial liver	Liver tissue engineering	Considerable mechanical strength, good hepatocyte compatibility as well as excellent blood compatibility	[123]
Chitosan/collagen/heparin matrix	Artificial liver	Aid in implantable bioartificial liver (IBL)applications	Superior blood compatibility	[123]
Chitosan modified with galactose	Artificial liver	Liver tissue engineering	Improve hepatocyte attachment and maintain viability	[124]
Galactosylated chitosan (GC)	Artificial liver	ECM for hepatocyte attachment	New synthetic ECM	[124]
Conjugated fructose onto the porous chitosan scaffold	Artificial liver	Liver tissue engineering	Induces cellular aggregates/enhances liver-specific metabolic activities and cell density	[120,122]
Chitosan	Artificial nerve	Neuron tissue culture	Repair of peripheral nervous system	[127,128]
Chitosan fibers	Artificial nerve	Regenerating axons to Büngner bands in the nervous system	Supported the adhesion, migration, and proliferation of SCs	[129]
Hydroxyapatite coated chitosan tubes loaded with laminin-1 or laminin peptides	Artificial nerve	Nerve regeneration	Peripheral nerve reconstruction	[130]
Immobilization of laminin peptide in molecularly aligned chitosan by covalent bonding	Artificial nerve	Nerve regeneration	Regeneration of PNS	[130,131]
Peptide conjugate chitosan membranes	Artificial nerve	Nerve repair	Accelerate axonal regeneration	[132]
Chitosan prostheses	Artificial nerve	Nerve regenerative response	Better facial response	[133]
Hexamethylene diisocyanate (HDI), epichlorohydrin (ECH) and glutaraldehyde (GA) as crosslinked chitosan films	Artificial nerve	Nerve regeneration	Enhanced the spread and proliferation of Schwann cells	[134]
Chitosan/poly(L-lysine) composite	Artificial nerve	Nerve regeneration	Improved nerve cell affinity	[135]
Chitosan/gelatin composite film	Artificial nerve	Nerve tissue culture	PC12 cells cultured on the composite films differentiated more rapidly and extended longer neurites	[136]
Chitin hydrogel tubes	Artificial nerve	Scaffolds in neural tissue engineering	Support nerve cell adhesion and neurite outgrowth	[137]
Complexation of GAGs with porous chitosan scaffolds	Artificial blood vessel	Vascular grafts	Growth inhibitory effects on vascular smooth muscle cells and their anticoagulant activity	[121]
Chitosan scaffolds, including heparin modified porous tubes	Artificial blood vessel	Blood vessel tissue engineering	Biocompatibility	[5]
Heparin–chitosan scaffolds	Artificial blood vessel	Blood vessel tissue engineering	Reduced coagulation, complement and blood cells. stimulates cell proliferation and formation of thick, vascularized granulation layer	[139,140,141,142,143,144]

## Data Availability

Not applicable.

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
