# Peer review of "Exploring the Impact of Chitosan Composites as Artificial Organs"

_polymers, 2022, doi:10.3390/polym14081587_

Round 1

Reviewer 1 Report

The short and concise review presents applicability of chitosan and its composites as biocompatible and biodegradable materials for tissue engineering and artificial organs fabrication. It touches current trends and will be definitely of interest for potential readers.

Description of chemical aspects of chitosan has been included in the light of biocompatibility, as well as its synthesis for biomedical applications. The described synthetic methods include solvent casting, compression molding, freeze drying and electrospinning compared to microwave assisted synthesis and dense gas foaming.

The snapshot of biomedical applications of chitosan is followed by a more detailed description of the use of this polymer in areas related to some specific artificial organs and tissue engineering. Chitosan can be used simply for the preparation of artificial membranes for dialysis. However a huge potential of the polymer in artificial organ regeneration is the main theme of the review. Chitosan-based materials can be not only used for skin tissue engineering and wound healing but also for the cell culture, reinforcement and repair of internal body parts such as bones, liver (tissue engineering as well as aids in the implanting process), cartilage and even nerve tissue (tissue culture and regeneration) and blood vessel tissue.

The review of application of chitosan towards artificial organs is focussed on the novel approach involving biocompatible and biodegradable materials in medicine. The overview of areas impacted by the biomaterial is followed by brief design of future perspectives and development.

I recommend publication of the manuscript.

Author Response

The short and concise review presents applicability of chitosan and its composites as biocompatible and biodegradable materials for tissue engineering and artificial organs fabrication. It touches current trends and will be definitely of interest for potential readers.

Description of chemical aspects of chitosan has been included in the light of biocompatibility, as well as its synthesis for biomedical applications. The described synthetic methods include solvent casting, compression molding, freeze drying and electrospinning compared to microwave assisted synthesis and dense gas foaming.

The snapshot of biomedical applications of chitosan is followed by a more detailed description of the use of this polymer in areas related to some specific artificial organs and tissue engineering. Chitosan can be used simply for the preparation of artificial membranes for dialysis. However a huge potential of the polymer in artificial organ regeneration is the main theme of the review. Chitosan-based materials can be not only used for skin tissue engineering and wound healing but also for the cell culture, reinforcement and repair of internal body parts such as bones, liver (tissue engineering as well as aids in the implanting process), cartilage and even nerve tissue (tissue culture and regeneration) and blood vessel tissue.

The review of application of chitosan towards artificial organs is focussed on the novel approach involving biocompatible and biodegradable materials in medicine. The overview of areas impacted by the biomaterial is followed by brief design of future perspectives and development.

I recommend publication of the manuscript.

Ans. Really grateful and motivated by your appreciation of our work. Thank you very much for your time and efforts.

Reviewer 2 Report

The article is very ambiguous and very general since chitosan is widely used for tissue engineering applications. The authors should focus on a specific topic, where chitosan is used...

Reviewer comments: It is still difficult to find the novelty of the work with respect to what has already been published. A literature review is required.

Reviewer comments: Several review articles have been published about "Chitosan for tissue engineering applications". What is the difference between what is published with what the authors want to publish? It is not clear. Also, the authors must describe all the review articles previously published related to the “Chitosan for tissue engineering applications” and describe their differences and novelty.

Reviewer comments: The number of new publications in the field is high and growing day-by-day. But it is also true that the number of (good) reviews in the area are equally increasing. Hence, I believe new reviews should be focused on the recent advances while making use of the efforts from previous reviews to substantiate the knowledge in the field.

Author Response

The article is very ambiguous and very general since chitosan is widely used for tissue engineering applications. The authors should focus on a specific topic, where chitosan is used...

Ans. Thank you for your valuable suggestions. As a review article we have focussed this article to highlight key areas in artificial organs. tissue engineering aspects have been widely reported, but not much has been reported in terms of artificial organs. this review takes up highlighting this specific area. Thank you. 

Reviewer comments: It is still difficult to find the novelty of the work with respect to what has already been published. A literature review is required.

Ans. We do understand your concern, we have presented a statistical graph now for clarity comparing, the number of studies involving chitosan and tissue engineering, chitosan and artificial organs and chitosan composites and artificial organs. As you will observe from the figure, chitosan composites and artificial organ applications are limited to less than 50 reports (42). This review specifically aims at highlighting and consolidating the chitosan composites and their applications into the area of artificial organs. While chitosan and tissue engineering aspects have been widely reviewed, there are only a handfull of reports on chitosan composites and artificial organs. We  agree that we have not highlighted the novelty of the review, we have now as per your recommendation, highlighted the novelty of this review. thank you. 

Reviewer comments: Several review articles have been published about "Chitosan for tissue engineering applications". What is the difference between what is published with what the authors want to publish? It is not clear. Also, the authors must describe all the review articles previously published related to the “Chitosan for tissue engineering applications” and describe their differences and novelty.

Ans. As explained above, we have now highlighted the uniqueness of the review, which was not that well brought out in our original version.  Chitosan for tissue engineering applications is a well reviewed subject area, our purpose is not that, we review a more narrower area within tissue engineering, specifically artificial organs, that has relatively amidst all other tissue engineering aspects been less reported. That is the uniqueness of the current review. We have elaborated on this in our revision. thank you. 

Reviewer comments: The number of new publications in the field is high and growing day-by-day. But it is also true that the number of (good) reviews in the area are equally increasing. Hence, I believe new reviews should be focused on the recent advances while making use of the efforts from previous reviews to substantiate the knowledge in the field.

Ans. Yes we completely understand your concern, we have updated and added all new information in this subject area in our revision. thank you again. 

Reviewer 3 Report

In general, reviews on the use of chitosan in tissue engineering are plentiful, and the quality of the presented review is inferior to a number of previous articles (https://doi.org/10.1155/2015/821279; https://doi.org/10.1016/j .ijbiomac.2018.04.034 and others). These and several other reviews should be cited by the authors and the novelty of the review presented should be assessed. In addition, it is necessary to add a literature search scheme to the article to review which databases were used, what are the statistics of articles on the use of chitosan in tissue engineering in recent years.

The authors should think further about what might become a feature of their review, which would make its existence necessary. In particular, it is necessary to add the most recent studies on chitosan and its composites as a material for tissue engineering (for example, https://doi.org/10.1016/j.carbpol.2021.118156; https://doi.org/10.1007/s42860- 021-00152-7; 10.1016/j.actbio.2017.06.001; https://doi.org/10.3390/polym13223949 and others).
In the review, insufficient attention is paid to the chemical features of chitosan, there are no formulas for modifications of chitosan, a comparison of the mechanical properties of pure chitosan and complex polymers, and ways to increase mechanical strength.
It is necessary to add figures or diagrams to each method for the preparation of biomaterials.
Line 143 - the name is incorrect. Probably, this refers to the method of forming materials based on chitosan.
In the section on the use of chitosan in the formation of organs and tissues, it is also necessary to add figures from original studies.

Author Response

In general, reviews on the use of chitosan in tissue engineering are plentiful, and the quality of the presented review is inferior to a number of previous articles (https://doi.org/10.1155/2015/821279; https://doi.org/10.1016/j .ijbiomac.2018.04.034 and others). These and several other reviews should be cited by the authors and the novelty of the review presented should be assessed. In addition, it is necessary to add a literature search scheme to the article to review which databases were used, what are the statistics of articles on the use of chitosan in tissue engineering in recent years.

Ans. We do understand your concern, the review you have pointed out, Chitosan scaffolds with enhanced mechanical strength and elastic response by combination of freeze gelation, photo-crosslinking and freeze-drying" is quiet different from the focus of our review.

However,we accept that the current manuscript does not adequately bring out the significance and the novelty of this work amidst others. As per your recommendation, we have now highlighted the focus and aim and novelty of this manuscript, in our revised version. we have also added a statistics on the reason why we chose to limit ourselves to a less worked on area, 'chitosan composites and artificial organs', instead of the routine chitosan and tissue engineering reviews. 

We have shown the comparative statistics of publications on chitosan and tissue engineering as well as chitosan composites and artificial organs and added this as a new figure in the manuscript. thank you

The authors should think further about what might become a feature of their review, which would make its existence necessary. In particular, it is necessary to add the most recent studies on chitosan and its composites as a material for tissue engineering (for example, https://doi.org/10.1016/j.carbpol.2021.118156; https://doi.org/10.1007/s42860- 021-00152-7; 10.1016/j.actbio.2017.06.001; https://doi.org/10.3390/polym13223949 and others).

Ans. We have cited  the related papers in the revision. thank you. 

In the review, insufficient attention is paid to the chemical features of chitosan, there are no formulas for modifications of chitosan, a comparison of the mechanical properties of pure chitosan and complex polymers, and ways to increase mechanical strength.
It is necessary to add figures or diagrams to each method for the preparation of biomaterials.

Ans. As you have addressed earlier, all these factors such as preparation methods, chemical formulae, properties of chitosan and complexes have all been reviewed and reported in numerous reviews, we have focused on highlighting aspects that have been less approached. and rather than diverging, we have concentrated on the theme of importance.  thank you for your understanding. 

Line 143 - the name is incorrect. Probably, this refers to the method of forming materials based on chitosan.

Ans. Sorry about that, we have now corrected this part. thank you for your patience. 

In the section on the use of chitosan in the formation of organs and tissues, it is also necessary to add figures from original studies.

Ans. We do not have access to such figures, the original publishers are entitled to authoritatively publish these. We stick to what we could free access. Thank you. We have added Fig 3 on the statistics of the current publication scenario on the areas related to this review. thank you very much for your patience. 

Round 2

Reviewer 2 Report

The article cannot be accepted in its present form.

Review comments: The review article is confusing. The authors describe that “This is the first review that consolidates the published literature related to chitosan composites and artificial organs”. However, the articles cited in each section only is related to specific properties of biomaterials for tissue engineering applications. I would expect to find references that demonstrate their functionality as synthetic organs... and not only describe biological properties such as biocompatibility, etc, etc.

Review comments: It is still difficult to find the novelty of the work concerning what has already been published. the authors must clearly describe the difference between what has already been published and what they want to do.

Review comments: The authors should discuss the challenges faced with the synthesis of artificial organs.

Review comments: The aim of the review article is not clear.

Review comments: The title of the review article is very general “Exploring the impact of chitosan and its composites as tissue analogs for artificial organs”. However, the number of articles that discuss the authors is very poor.

Review comments: The authors need to be updated with more recent references since the topic is quite reported.

Review comments: Anthony Atala is a pioneer in the synthesis and functionality of artificial organs. However, the authors do not seem to know about it. I would expect that the main articles of this important researcher would be discussed in this review.

Review comments: The authors must describe the main physicochemical and mechanical properties that biomaterials must possess so that they can mimic the organs described.

Author Response

Review comments: The review article is confusing. The authors described that “This is the first review that consolidates the published literature related to chitosan composites and artificial organs”. However, the articles cited in each section only is related to specific properties of biomaterials for tissue engineering applications. I would expect to find references that demonstrate their functionality as synthetic organs... and not only describe biological properties such as biocompatibility, etc, etc.

Ans. We do understand your point, whatever we have mentioned in section 4 is all that is available with respect to chitosan composites and artificial organs. We do agree with you that most of these are not direct synthetic organ construction applications. This is something we are trying to bring out in the review; to prompt more high-level accelerated implementation of chitosan composites to aspects concerning artificial organs.   We have however, done a complete resweep and added anything that we left out. Thank you.                

Review comments: It is still difficult to find the novelty of the work concerning what has already been published. The authors must clearly describe the difference between what has already been published and what they want to do.

Ans.  The novelty of the work is that even in terms of the title, chitosan composites and artificial organs, there are no other articles. We review this area and summarize the available work and highlight the areas that need more inputs. This is the novelty of the review. Thank you.    

Review comments: The authors should discuss the challenges faced with the synthesis of artificial organs.

Ans. Synthesis of artificial organs is an entirely wide subject, that has its own side of reviews. We focus on the use of chitosan composites for synthesis of artificial organs and discuss the challenges in this aspect. Thank you.  

Review comments: The aim of the review article is not clear. Review comments: The title of the review article is very general “Exploring the impact of chitosan and its composites as tissue analogs for artificial organs”. However, the number of articles that discuss the authors is very poor.

Ans. We have once again revised and explained the aim of the article. We have also modified the title and made it more specific.

Review comments: The authors need to be updated with more recent references since the topic is quite reported.

Ans.We have updated the references as what exists. Thank you.

Review comments: Anthony Atala is a pioneer in the synthesis and functionality of artificial organs. However, the authors do not seem to know about it. I would expect that the main articles of this important researcher would be discussed in this review.

Ans.We have added a note. Thank you.

Review comments: The authors must describe the main physicochemical and mechanical properties that biomaterials must possess so that they can mimic the organs described.

Ans. Added a brief on this in the revised version.

Reviewer 3 Report

The authors generally took into account all the comments.

Author Response

The authors generally took into account all the comments.

Ans. Thank you dear reviewer for your consideration and acceptance of our article

Thank you for your time and valuable inputs.

Round 3

Reviewer 2 Report

The article can be accepted